# A Reproducibility Study of Decoupling Feature Extraction and Classification Layers for Calibrated Neural Networks

**Eric Banzuzi** *banzuzi@kth.se*
*KTH Royal Institute of Technology*

**Johanna D'Ciofalo Khodaverdian** *jdk@kth.se*
*KTH Royal Institute of Technology*

**Katharina Deckenbach** *kadec@kth.se*
*KTH Royal Institute of Technology*

**Reviewed on OpenReview:** *https://openreview.net/forum?id=5Hwzd48ILf*

## Abstract

Many neural networks, especially over-parameterized ones, suffer from poor calibration and overconfidence. To address this, Jordahn & Olmos (2024) recently proposed a Two-Stage Training (TST) procedure that decouples the training of feature extraction and classification layers. In this study, we replicate their findings and extend their work through a series of ablation studies. We reproduce their main results and find that most of them replicate, with slight deviation for CIFAR100. Additionally, we extend the author's results by exploring the impact of different model architectures, Monte Carlo (MC) sample sizes, and classification head designs. We further compare the method with focal loss – an implicit regularization technique known to improve calibration – and investigate whether calibration can be improved further by combining the two methods. Beyond focal loss, we also evaluate the effect of incorporating other similar regularization techniques such as label smoothing and L2 regularization during two-stage training. We find that calibration can be improved even further by using focal loss in the first training stage of two-stage training. Similar improvements are observed when combining two-stage training with label smoothing and L2 regularization. Our experiments validate the claims made by Jordahn & Olmos (2024), and show the transferability of the two-stage training to different architectures.

## 1 Introduction

Calibration is crucial for trustworthy machine learning, especially in safety-critical fields like medicine and self-driving cars, where understanding a model's uncertainty is key to assessing decision reliability. Neural networks, particularly over-parameterized ones, tend to be poorly calibrated and have an overconfidence bias. Efforts to improve calibration include post-hoc methods and modifications to training procedures (Wang, 2023).

In *Decoupling Feature Extraction and Classification Layers for Calibrated Neural Networks*, Jordahn & Olmos (2024) propose Two-Stage Training (TST) and Variational Two-Stage Training (V-TST), for calibration of Deep Neural Networks (DNNs). The methods decouple the training of the feature extraction and classification layers. At first, all layers are trained jointly from scratch and in the second stage of training, the last layer is re-initialized and re-trained from scratch while the weights of the feature extraction layers are kept frozen.

We conducted several experiments to analyze the performance of TST and V-TST under different configurations, and extend their method further by combining it with focal loss – both its constant and adaptive variant. Our main contributions are:

- Reproducing the main results (first two rows of Tables 1 and 2 of Jordahn & Olmos (2024)).

- Reproducing two of their ablation studies, namely applying their method to an under-parameterized CNN and a fine-tuned ViT (Tables 3 and 6 of Jordahn & Olmos (2024)).

- Conducting three ablation studies to investigate the dependence on model architecture, the sample size of a Monte Carlo (MC) estimation used in the second-stage of training and the architecture of the classification head.

- Further extending the original paper by comparing and combining the method with three different regularization techniques, namely focal loss (Lin et al., 2017), label smoothing (Szegedy et al., 2016) and L2 regularization.

## 2  Related Work

Neural network calibration methods can be grouped into four different categories (Wang, 2023; Gawlikowski et al., 2023): post-hoc calibration, regularization methods, uncertainty estimation, and methods that combine the aforementioned categories. Within the category of post-hoc methods, Temperature Scaling (TS) (Guo et al., 2017) is a widely used, simple and computationally efficient approach with many variations (Wang, 2023). Post-hoc methods are generally characterized by their applicability after training. While the main goal of uncertainty estimation approaches is to provide uncertainty estimates of model predictions, they can also improve calibration by reducing epistemic uncertainty (Gawlikowski et al., 2023). Prominent examples of these techniques are and ensemble methods and Bayesian neural networks (BNNs). Regularization techniques encompass data augmentation (Thulasidasan et al., 2019), label smoothing (LS) (Szegedy et al., 2016), L2 regularization Guo et al. (2017), exposure to OOD samples (Hendrycks et al., 2019), objective function modification (Pereyra et al., 2017), methods that use calibration errors as differentiable proxy (Kumar et al., 2018; Wang, 2023) as well as the use of different loss functions (Hebbalaguppe et al., 2022; Shamsi et al., 2023; Lin et al., 2017).

For this study, implicit regularization techniques are the most relevant as TST and V-TST fall into this category. An example of such an implicit technique is Focal Loss (Lin et al., 2017), defined as

$$\text{FL}(p_t) = -(1 - p_t)^\gamma \log\,(p_t) \tag{1}$$

Focal loss is a modification of cross-entropy loss designed to address class imbalance during training. It introduces a weighting factor, $(1 - p_t)^\gamma$, where $p_t$ is the predicted probability of the true class. This factor down-weights the loss of well-classified examples ($p_t \approx 1$) and amplifies the influence of hard examples ($p_t \approx 0$), making training more effective. The focusing parameter $\gamma \geq 0$ determines the degree of this effect. Focal loss has been shown to be an upper bound on the regularized KL-divergence and improves calibration of networks independent of their architecture (see Mukhoti et al. (2020) for a more elaborate discussion on why focal loss improves calibration). The success of focal loss has been connected to its ability to control the curvature of the loss landscape (Kimura & Naganuma, 2025). Several advancements of focal loss have been suggested, such as dual focal loss (Tao et al., 2023a), calibration-aware adaptive focal loss called AdaFocal (Ghosh et al., 2022), or combining focal loss with temperature scaling (Komisarenko & Kull, 2024).

Another widely used implicit regularization technique is label smoothing (Szegedy et al., 2016), which reduces the confidence of predictions by computing the cross-entropy loss with a smoothened target distribution that assigns a small portion of the probability mass to incorrect classes. Instead of assigning the true probability to the target class, label smoothing assigns a slightly lower probability $1 - \epsilon$ to the target class and distributes the remaining $\epsilon$ uniformly across the incorrect classes. Similarly to focal loss, label smoothing regularization reduces overconfidence and generally improves model calibration (Müller et al., 2019).

## 3  Methods

Jordahn & Olmos (2024) propose a novel two-stage training procedure for improved calibration of DNNs in classification tasks. Typically, the training of feature extraction layers and classification layers occurs

jointly. Their findings indicate that training these layers separately in a two-stage fashion can improve model calibration. They argue that freezing the feature extraction layers limits the model's flexibility, preventing the classifier from artificially increasing label likelihood. Thus, their method falls in the category of regularization techniques, and the success of their method is connected to finding the perfect regularization for a learned feature extractor (Jordahn, 2024). In their paper, they present two variations of the two-stage training method: TST and V-TST.

### 3.1 TST

In TST, a model with initial parameters $\{\beta, \phi\}$, denoting the parameters of the feature extraction layers and classification layers respectively, is trained from scratch using cross-entropy loss (Jordahn & Olmos, 2024). The parameters of the feature extraction layers $\beta$ are then frozen and the classification layers are re-initialized and re-trained from scratch using the same dataset $D_{\text{train}}$ (see Fig.1). Further calibration improvements can be obtained by adjusting the dimensionality $Z$ of the last hidden layer $z$ of the classification layers, as the calibration performance varies with different choices of $Z$.

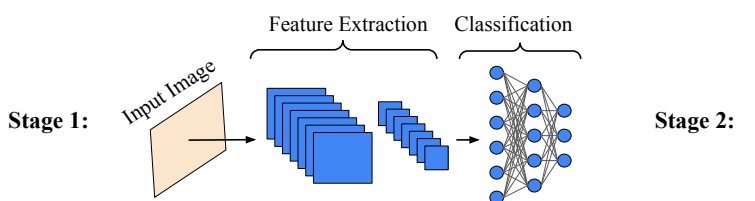
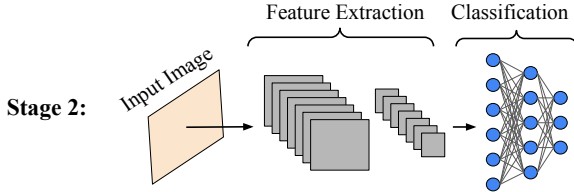

(a) First training stage: feature extraction and classification layers are trained jointly from scratch.

(b) Second training stage: feature extraction layers are frozen and classification layers are re-initialized and trained from scratch.

Figure 1: Illustration of training procedure for TST and V-TST. Blue indicates the layers are being trained while gray indicates frozen layers.

### 3.2 V-TST

V-TST poses additional regularization on $z$. For the second stage, the parameters of the fully connected (FC) layers are optimized using the evidence lower-bound (ELBO):

$$\log p(\boldsymbol{y}) \geq \text{ELBO}_{\theta,\nu} = \mathbb{E}_{q(\boldsymbol{z}|\boldsymbol{x})}\log[p(y|\boldsymbol{z})] - D_{KL}(q(\boldsymbol{z}|\boldsymbol{x})||p(\boldsymbol{z})). \tag{2}$$

They assume a standard normal Gaussian prior $p(\boldsymbol{z}) = \mathcal{N}(\boldsymbol{0}, \boldsymbol{I})$ and an approximate posterior of the form $q(\boldsymbol{z}|\boldsymbol{x}) = \mathcal{N}(\mu_{\beta,\theta}(\boldsymbol{x}), \sigma_{\beta,\theta}(\boldsymbol{x}))$. The predicted output label of the model is given by $p(y|\boldsymbol{z}) = \text{Cat}(\pi_\nu(\boldsymbol{z}))$. To approximate and sample from $q(\boldsymbol{z}|\boldsymbol{x})$, they apply the reparameterization trick $\boldsymbol{z} = \mu_{\beta,\theta}(\boldsymbol{x}) + \sigma_{\beta,\theta}(\boldsymbol{x}) \odot \epsilon$ with $\epsilon \sim \mathcal{N}(\boldsymbol{0}, \boldsymbol{I})$. This allows the expectation of $p(y|\boldsymbol{z})$ to be determined by MC sampling using $m$ samples, as described in Kingma & Welling (2022).

### 3.3 Reproducing results

The suggested two-stage training methods are notable because they can be applied to pretrained networks. As such, they are neither a strict post-hoc nor an online-learning approach like data augmentation and can potentially be combined with other calibration methods. Another advantage is that they impose less additional computational cost than training an ensemble. Replacing the last layer with a MLP is applicable to a wide range of architectures, and during the second stage of training, models converge quite fast, as Jordahn & Olmos (2024) pointed out. Furthermore, their method outperformed Temperature Scaling, which is another popular post-hoc calibration method. These results look promising, even though their technique was not derived theoretically. In fact, they discovered this method when conducting other experiments (Jordahn, 2024). Therefore, a further investigation of their observed phenomenon can contribute to a better

understanding. While this paper focuses on the empirical side, we hope to encourage a thorough theoretical analysis.

Using the code provided by the authors, we trained a WRN-28-10 architecture from scratch to reproduce the main results. We used the proposed methods TST and V-TST on the CIFAR10 and CIFAR100 datasets and included the evaluation on the shifted and out-of-distribution (OOD) data. We also reproduced two of the ablation studies in the original paper. The first ablation study evaluates the performance of two-stage training on an under-parameterized CNN architecture. Secondly, we performed the ablation study regarding two-stage training for pre-trained models using ViT-B/16 (Dosovitskiy et al., 2021) on Tiny ImageNet (Le & Yang, 2015a).

### 3.4 Our extensions

In addition to reproducing the original results, we introduced several extensions to deepen our understanding of TST and V-TST, and to examine how various experimental setups influence their performance. While Jordahn & Olmos (2024) conducted a range of calibration studies, such as exploring the effects of end-to-end training, we aim to build on this by performing new ablation studies within a similar scope to gain new insights. For all experiments, we extend the results from Jordahn & Olmos (2024) by also including the Brier score to bring new insights. See Appendix B for a detailed motivation of including the Brier score on top of ECE and MCE.

First, we investigated the effectiveness of two-stage training using two additional CNN architectures: ResNet (He et al., 2015) and EfficientNet (Tan & Le, 2020). ResNet was chosen since it serves as a canonical baseline for many subsequent CNN designs, making it a natural choice for comparison. EfficientNet was selected due to its superior performance on benchmarks like ImageNet, which makes it an appealing architecture for practical deployment Tan & Le (2020). Moreover, EfficientNet differs more substantially from WRN-28-10 by incorporating features such as compound scaling (Tan & Le, 2020), making it particularly interesting to assess the generalizability of two-stage training. Specifically, we used ResNet-50 (He et al., 2015) and EfficientNet-B5 (Tan & Le, 2020), which are comparable in size to WRN-28-10 (36.5 million parameters), with 25.6 million and 30.4 million parameters, respectively.

Next, we investigated different MLP configurations in the TST and V-TST setting, and examined the effect of varying the number of MC samples used in the second stage of V-TST. While the original paper used a three-layer MLP for the classification layer, we experimented with replacing it with a four-layer MLP and a single-layer MLP, the latter being equivalent to re-initializing the final fully connected (FC) layer. Additionally, we explored whether increasing the number of MC samples in the second stage with V-TST from $m = 1$ (original proposed approach) to $m = 5$ could improve model calibration. These ablation studies were motivated by the observation that the choice of the latent dimension $Z$ has a large effect on the calibration of the model and the effectiveness of two-stage training, as reported by Jordahn & Olmos (2024) in appendix E. For instance, using $Z = 128$ compared to $Z = 8$ in V-TST ($m = 10$) results in an ECE value of 1.270 instead of 12.88. We wanted to investigate whether the choice of other hyperparameters has a similarly strong impact on the performance of two-stage training.

Lastly, as suggested in Jordahn & Olmos (2024), we investigated whether combining two-stage training with focal loss and other similar regularization methods can lead to further improvements in calibration performance. Jordahn & Olmos (2024) suggested that their method is complementary rather than an alternative to other implicit regularization techniques. For a thorough evaluation of their claim, we explored whether calibration improvements can be seen when combining two-stage training with two implicit regularization techniques (focal loss and label smoothing) and one explicit regularization method (L2 regularization). We leave the combination of their method with last-layer Laplace, OKO, and similar techniques for future studies.

## 4 Data

We used the same image datasets and preprocessing as the original paper, namely CIFAR10, CIFAR100 (Krizhevsky et al., 2009) and Tiny ImageNet (Le & Yang, 2015b) to train the models, and SVHN (Netzer

et al., 2011) (OOD data) as well as CIFAR10-C and CIFAR100-C (Hendrycks & Dietterich, 2019)(shift data) to evaluate them. We randomly split the validation set of Tiny ImageNet into two halves to obtain validation and test sets, as the test set of the dataset is unlabeled. More details about the datasets and preprocessing steps can be found in Appendix A. For the ablation studies and the extensions, we used CIFAR10.

## 5 Experiments and Findings

Across all experiments, we evaluated the model performance using the metrics of the original paper, the expected calibration error (ECE) and the maximum calibration error (MCE) (Naeini et al., 2015), computed in percentage with 10 bins, on the regular and corrupted test sets. Furthermore, we measured negative log-likelihood (NLL) on both the training set and the test set, and included metrics AUROC for the shifted datasets and false-positive rate at 95% (FPR95) for the OOD datasets. In addition to the metrics used by the authors of the original paper, we included the Brier score, which is the mean squared error (MSE) applied to predicted probabilities, to get a more comprehensive understanding. As in the original paper, we used the softmax outputs of the trained models as inputs for computing the evaluation metrics. Note, that treating the softmax output of networks as their confidence has been criticized (Gal & Ghahramani, 2016). The softmax operation squeezes the prediction probability towards extreme values such as zero or one (He et al., 2025) which could cause the observed miscalibration (Gawlikowski et al., 2023; Vasudevan et al., 2019; Hendrycks & Gimpel, 2018). However, as our main objective is to reproduce the results of Jordahn & Olmos (2024), we decided to follow their experimental settings closely.

In all experiments, the models were trained using the Adam optimizer with learning rate $10^{-4}$ and cross-entropy (CE) loss to align with the original paper, except in the extensions where we used focal loss (FL). In the first stage, the models were trained from scratch for 600 epochs with seed 1 unless otherwise specified. Throughout the paper, we refer to models after the first stage of training as *base models*. The TST and V-TST was done over 10 seeds (0-9) using 40 epochs and by re-initializing the final FC layer with a three-layer MLP head having dimensions [`output`, $3Z$], [$3Z$, $Z$] and [$Z$, #classes], where the `output` corresponds to the dimension of the model architecture's output before the classifier. For each experiment, we followed the approach of the original code and stored the model with with the lowest validation loss. A single NVIDIA T4 16GB GPU was used for all experiments.

### 5.1 Replication of Main Results

The replication of the main results is reported in Tables 1 and 2. For TST and V-TST, we used the same latent dimensions $Z$ as the original paper. The CIFAR10 results for WRN-28-10 closely align with those reported in Jordahn & Olmos (2024). Especially the NLL results are very similar and are mostly consistent with the Brier results. No single model performs best on all metrics, however, V-TST ($m = 10$) has the best overall performance. The numbers are generally slightly worse for almost all metrics but lie within a close range to those found in the original paper. We believe that these slight differences can be attributed to the randomness involved in training the base model, leading to different base models to start from. While the authors specify running TST and V-TST over seeds 0-9, they do not specify the seed used for training the base model.

For CIFAR100, the results we obtained differ slightly from those reported by Jordahn & Olmos (2024). First of all, we observed discrepancies in the base WRN-28-10 models. Despite trying eight different seeds, we were unable to replicate comparable results. Our base model consistently showed lower accuracy, which, in contrast to the original results, increased by a few percentage points during the second stage of training. A direct comparison of ECE and MCE for our and their base model is however difficult, as the bias of these metrics depends on the accuracy (Minderer et al., 2021). Please refer to Appendix B for more details about our replication attempts and a short discussion of the dependence of ECE and MCE on the accuracy.

As the accuracy of our WRN-28-10 model trained on CIFAR100 differed by up to several percentage points, one should be careful when drawing conclusions about the ECE and MCE differences obtained. We considered the Brier score and NLL in addition and found that for our models, the Brier scores aligned with the NLL results. Similar to the findings in Jordahn & Olmos (2024), we observed improvements in both calibration and

accuracy when applying TST and V-TST, although the magnitude of these improvements differed compared to the ones of the original paper. For instance, V-TST ($m = 10$) improved the ECE by 60.85%, whereas the original paper reported a 73.0% improvement. A similar trend was observed for TST. Taking into account all metrics, V-TST ($m = 10$) seems best calibrated.

Consistent with Jordahn & Olmos (2024), we showed that V-TST and TST improve ECE and MCE across both datasets (see Table 1). Under distribution shifts and OOD settings, we similarly found that TST and V-TST significantly improve the calibration of the WRN-28-10 and outperform temperature scaling for CIFAR10 and CIFAR100.

Table 1: Replication of main results. Test metrics on in-distribution test datasets. For V-TST, $m = 1$ and $m = 10$ indicate using 1 and 10 samples in the MC approximation, respectively. Temp. WRN is the temperature-scaled WRN. For CIFAR10, we used $Z = 128$ for V-TST ($m = 10$) and TST, and $Z = 32$ for V-TST ($m = 1$). For CIFAR100, $Z = 512$ was used for V-TST ($m = 10$) and TST, and $Z = 128$ for V-TST ($m = 1$). Bold indicates the best-performing model.

| Dataset | Model | Accuracy | ECE | MCE | Train NLL | Test NLL | Brier |
|---|---|---|---|---|---|---|---|
| CIFAR10 | V-TST, $m = 10$ | 92.44±0.02 | **1.38±0.100** | **21.26±6.44** | 0.0655±0.0004 | 0.2579±0.0017 | **0.0114±0.00004** |
| | V-TST, $m = 1$ | 90.9±0.03 | 1.78±0.148 | 24.80±8.981 | 0.1116±0.0012 | 0.3732±0.0041 | 0.0145±0.00007 |
| | TST | **92.55±0.01** | 3.34±0.016 | 24.68±3.883 | **0.0532±0.0002** | **0.2516±0.0007** | 0.0115±0.00001 |
| | Temp. WRN | 92.52 | 4.70 | 25.2 | 0.0616 | 0.3024 | 0.0122 |
| | WRN | 92.52 | 6.06 | 36.28 | 0.1031 | 0.5124 | 0.0133 |
| CIFAR100 | V-TST, $m = 10$ | **68.29±0.06** | **9.33±0.227** | **19.75±0.611** | **0.140±0.0025** | **1.3551±0.005** | **0.0045±0.00001** |
| | V-TST, $m = 1$ | 66.30±0.05 | 10.60±0.391 | 21.12±0.820 | 0.187±0.0027 | 1.5510±0.0108 | 0.0048±0.00001 |
| | TST | 66.91±0.03 | 9.96±0.042 | 18.81±0.305 | 0.1921±0.0005 | 1.3530±0.0014 | 0.0046±0.000005 |
| | Temp. WRN | 62.63 | 12.07 | 22.42 | 0.3272 | 1.5512 | 0.0052 |
| | WRN | 62.63 | 23.83 | 46.73 | 0.3309 | 2.1619 | 0.0059 |
| CIFAR10 original | V-TST, $m = 10$ | **92.58±0.02** | **1.27±0.101** | 28.48±9.006 | 0.0671±0.0004 | 0.2617±0.0013 | |
| | V-TST, $m = 1$ | 91.09±0.06 | 1.52±0.144 | **14.77± 1.781** | 0.1125±0.001 | 0.3732±0.0043 | |
| | TST | **92.59±0.02** | 2.18±0.353 | 14.89±2.249 | **0.0573±0.0002** | **0.2482±0.004** | |
| | Temp. WRN | 92.53 | 4.4 | 26.77 | 0.063 | 0.2965 | |
| | WRN | 92.53 | 5.88 | 35.84 | 0.1038 | 0.4944 | |
| CIFAR100 original | V-TST, $m = 10$ | **71.93±0.04** | **5.83±0.143** | **14.03±0.59** | 0.078±0.001 | 1.202±0.0052 | |
| | V-TST, $m = 1$ | 69.18±0.09 | 7.34±0.291 | 16.85±0.694 | 0.1094±0.0009 | 1.4315±0.0104 | |
| | TST | 71.26±0.06 | 7.07±0.084 | 16.74±1.942 | 0.0887±0.0003 | **1.1331±0.002** | |
| | Temp. WRN | 71.55 | 15.24 | 33.21 | **0.0739** | 1.3708 | |
| | WRN | 71.56 | 21.74 | 82.08 | 0.1161 | 2.2588 | |

### 5.1.1 Replication of CNN ablation study

To reproduce Table 3, we implemented a simple CNN, following the architecture detailed in the Appendix of the original paper. After training the base model, we applied TST and V-TST and matched the latent dimension with their results. Consistent with Jordahn & Olmos (2024), we found that two-stage training on an under-parameterized model worsens ECE but improves MCE, while TST can improve both as shown in Table 17 (Appendix E). The lower proportion of frozen to trainable parameters in the second stage of an under-parameterized model increases flexibility, possibly explaining the ineffectiveness of two-stage training. Additionally, we observed that for TST, the authors report improved ECE with $Z = 32$, the only latent dimension where ECE improved. With $Z = 128$, we found the ECE to be worse than the base model.

### 5.1.2 Replication of ViT ablation study

The other ablation study concerned applying the two-stage training after fine-tuning a pre-trained ViT. Implementation details are described in Appendix C. Again, our results were generally consistent with their findings. It can be seen from Table 18 (Appendix E) that using V-TST can improve the ECE and MCE while TST worsens both ECE and MCE. They stress the good regularization properties of V-TST and also conclude that training feature extraction layers and classification layers together from scratch is important for the two-stage training to be effective.

Table 2: Replication of main results. Evaluation metrics on shifted data and OOD data.

| Dataset | Model | SHIFT ECE | SHIFT MCE | OOD AUROC | OOD FPR95 |
|---------|-------|-----------|-----------|-----------|-----------|
| CIFAR10 | V-TST, $m = 10$ | **9.41±0.3** | 20.33±0.66 | 0.851±0.003 | 0.702±0.002 |
| | V-TST, $m = 1$ | 12.53±0.29 | **20.19±0.56** | 0.766±0.003 | 0.778±0.003 |
| | TST | 13.09±0.04 | 29.32±0.10 | 0.883±0.001 | **0.688±0.003** |
| | Temp. WRN | 16.44 | 37.93 | **0.885** | 0.698 |
| | WRN | 20.06 | 45.62 | 0.874 | 0.708 |
| CIFAR100 | V-TST, $m = 10$ | **18.99±0.25** | 32.52± 0.46 | **0.784±0.001** | **0.783±0.003** |
| | V-TST, $m = 1$ | 20.81±0.48 | 35.53±0.70 | 0.770±0.001 | 0.803±0.003 |
| | TST | 21.13±0.76 | **30.82±1.13** | 0.692±0.002 | 0.846±0.004 |
| | Temp. WRN | 23.67 | 37.71 | 0.744 | 0.840 |
| | WRN | 39.41 | 56.06 | 0.722 | 0.854 |
| CIFAR10 original | V-TST, $m = 10$ | **10.41±0.22** | 22.92±0.57 | 0.821±0.003 | 0.751±0.002 |
| | V-TST, $m = 1$ | 12.89±0.3 | **21.01±0.58** | 0.722±0.005 | 0.824±0.006 |
| | TST | 11.62±0.75 | 25.39±1.74 | **0.874±0.002** | 0.699±0.009 |
| | Temp. WRN | 16.45 | 36.77 | 0.872 | 0.746 |
| | WRN | 20.27 | 45.06 | **0.891** | **0.653** |
| CIFAR100 original | V-TST, $m = 10$ | **14.3±0.23** | **26.96±0.46** | 0.791±0.002 | 0.809±0.004 |
| | V-TST, $m = 1$ | 16.8±0.4 | 30.72±0.63 | 0.783±0.004 | 0.822±0.006 |
| | TST | 17.44±0.08 | 28.51±0.14 | **0.823±0.002** | **0.764±0.006** |
| | Temp. WRN | 29.58 | 47.67 | 0.778 | 0.816 |
| | WRN | 40.47 | 60.38 | 0.785 | 0.892 |

## 5.2 Ablation Study: Dependence on Network Architecture

As part our ablation studies, we investigated whether TST and V-TST generalize to other CNN architectures, specifically ResNet-50 (He et al., 2015) and EfficientNet-B5 (Tan & Le, 2020) (see Section 3.4 for the motivation behind these choices). To accommodate the smaller $32 \times 32$ input size of CIFAR10, we adapted the architectures of ResNet-50 and EfficientNet-B5 (see Appendix C for details). All models were trained from scratch before applying TST and V-TST.

Both TST and V-TST improved calibration for ResNet-50 and EfficientNet-B5 (see Table 3). The best-performing model for ResNet-50 was obtained with V-TST ($m = 10$), which aligns with the results for WRN-28-10. V-TST ($m = 10$) improved ECE and MCE by 88.14% and 62.17%, respectively. This model also reached the highest accuracy of 89.80%. For EfficientNet-B5, the best calibration was obtained for temperature scaling, which improved ECE and MCE by 90.74% and 92.50%, respectively. The best-performing two-stage procedure for EfficientNet-B5 was V-TST ($m = 1$), for which ECE and MCE was improved by 40.93% and 81.42%.

Calibration also improved when using TST and V-TST on shifted data. For ResNet-50, the best performance was obtained for V-TST ($m = 10$), while temperature scaling outperforms two-stage training for EfficientNet-B5. For both architectures, the model with the lowest Brier score was also the best-performing model across all of the calibration metrics for in-distribution and shifted data. However, this tendency did not hold for OOD data. Furthermore, the OOD data detection performance decreased for ResNet-50 when applying two-stage training.

The similarity in performance trends between WRN-28-10 and ResNet-50 may stem from their architectural resemblance, as WRN builds upon the ResNet design by increasing the width of residual blocks while reducing depth (Zagoruyko & Komodakis, 2017). In contrast, EfficientNet differs more from WRN-28-10, by using features like compound scaling (Tan & Le, 2020). These architectural differences may explain why the results achieved with WRN-28-10 did not directly translate to EfficientNet. Exploring alternative latent dimensions could further enhance the performance of TST and V-TST for models like EfficientNet.

In summary, TST and V-TST seem to improve calibration across multiple CNN architectures. However, the extent of improvement appears to be architecture-dependent. For ResNet-50, that architecturally closely relates to WRN-28-10, V-TST remains the most effective approach. In contrast, for EfficientNet-B5 whose

architecture diverge more significantly, simpler post-hoc methods such as temperature scaling outperform two-stage training. This highlights the importance of considering architectural characteristics when selecting calibration strategies.

Table 3: Dependence of TST and V-TST on network architecture. The latent dimensions used for TST, V-TST ($m = 1$), and V-TST ($m = 10$) were the same as those used for WRN-28-10, namely $Z = 128$, $Z = 32$ and $Z = 128$, respectively.

| Model | Accuracy | ECE | MCE | Brier | SHIFT ECE | SHIFT MCE | OOD AUROC | OOD FPR95 |
|---|---|---|---|---|---|---|---|---|
| V-TST, $m = 10$ | **89.80±0.03** | **0.95±0.081** | **13.15±2.07** | **0.0154±0.00004** | **6.36±0.24** | **11.17±0.38** | 0.776±0.004 | 0.810±0.003 |
| V-TST, $m = 1$ | 86.77±0.09 | 1.60±0.075 | 30.52±6.829 | 0.0204±0.0001 | 10.95±0.40 | 15.40±0.63 | 0.719±0.001 | 0.852±0.002 |
| TST | 89.77 ±0.02 | 4.19±0.016 | 18.25±1.670 | 0.0156±0.00003 | 13.38±0.06 | 25.94±0.10 | 0.845±0.001 | 0.753±0.004 |
| Temp. ResNet-50 | 89.74 | 5.82 | 22.09 | 0.0164 | 16.83 | 33.82 | **0.861** | **0.744** |
| ResNet-50 | 89.74 | 8.01 | 34.75 | 0.0236 | 21.58 | 44.94 | 0.852 | 0.762 |
| V-TST, $m = 10$ | 76.93±0.04 | 4.97±0.267 | 16.40±1.926 | 0.0327±0.0005 | 13.90±0.34 | 20.73±0.55 | 0.683±0.001 | **0.905±0.001** |
| V-TST, $m = 1$ | 74.87±0.08 | 4.85±0.270 | 14.99±1.615 | 0.0364±0.0001 | 14.54±0.26 | 19.91±0.36 | 0.663±0.001 | 0.906±0.002 |
| TST | **77.10±0.04** | 7.20±0.225 | 43.68±8.575 | 0.0330±0.00004 | 16.54±0.30 | 25.43±0.47 | **0.689±0.001** | 0.909±0.001 |
| Temp. EfficientNet-B5 | 76.48 | **0.76** | **6.05** | **0.0302** | **8.95** | **12.80** | 0.681 | 0.920 |
| EfficientNet-B5 | 76.48 | 8.21 | 80.68 | 0.0315 | 18.58 | 28.35 | 0.678 | 0.917 |

## 5.3 Ablation Study: Second Stage Network Architecture

In order to investigate how the complexity of the MLP head architecture in the second stage of TST and V-TST influences the model's calibration, we conducted experiments by varying the original three-layer MLP head to a single-layer and a four-layer MLP, while keeping the latent dimensions consistent with the replication experiments in section 5.1 across all setups. For further details, we refer to Appendix C.

The results in Tables 4 and 5 show that TST and V-TST with a four-layer MLP improved ECE and MCE with and without distribution shifts. While the performance is close to that of the three-layer MLP, it is slightly worse overall when compared to all metrics. For TST, the lower train NLL and higher test NLL show signs of more overfitting in the four-layer configuration. We believe that the additional layer without optimizing the dimension of $Z$ introduces too many trainable parameters. We also noticed that the models converged faster with four layers. Furthermore, the results of TST using a four-layered MLP but with a decreased latent dimension of $Z = 10$ show comparable results to the ones with a three layer MLP and latent dimension of $Z = 128$, highlighting the potential importance of balancing number of layers in the MLP and the latent dimension.

In the single-layer configuration, the results show that re-training the final FC layer in TST improves the MCE by 49.37% and the overall performance across the shift and OOD metrics, still outperforming temperature scaling in most metrics. This indicates that simply re-training the FC layer can effectively enhance model calibration and robustness under distribution shifts. However, the single-layer MLP performs worse compared to its multi-layer counterparts in terms of ECE and MCE on in-distribution data, likely due to the reduced capacity to capture complex calibration patterns. It is interesting to see that even though the results for this model seem consistently promising across NLL, Brier score, OOD data, and SHIFT ECE, this is not reflected in its ECE, MCE, and SHIFT MCE.

Table 4: Results for size of the MLP in the second stage of trainingon in-distribution data. The value $l$ indicates the number of layers in the MLP head of TST and V-TST. We used $Z = 128$ as latent dimension unless stated otherwise.

| Model | Accuracy | ECE | MCE | Train NLL | Test NLL | Brier |
|---|---|---|---|---|---|---|
| V-TST, $l = 4$, $m = 10$ | 92.50±0.02 | 1.95±0.141 | 26.08±5.912 | 0.0733±0.0006 | 0.2823±0.0029 | 0.0118±0.00005 |
| V-TST, $l = 4$, $m = 1$ | 91.65±0.05 | **1.71±0.029** | 27.94±8.612 | 0.1227±0.0011 | 0.3971±0.0026 | 0.0141±0.00006 |
| TST, $l = 4$ | **92.52±0.02** | 3.50±0.280 | 23.36±2.717 | 0.0547±0.0002 | 0.2588±0.0010 | 0.0115±0.00002 |
| TST, $l = 4$, $Z = 10$ | 91.64±0.11 | 2.45±0.150 | **14.74±0.997** | 0.0702±0.0021 | 0.3052±0.0091 | 0.0125±0.00017 |
| TST, $l = 1$ | **92.52±0.02** | 2.69±1.203 | 18.37±2.763 | **0.0526±0.00001** | **0.2408±0.0016** | **0.0113±0.00003** |

Table 5: Results for size of the MLP in the second stage of training on shifted and OOD data. The value $l$ indicates the number of layers in the MLP head of TST and V-TST. We used $Z = 128$ as latent dimension unless stated otherwise.

| Model | SHIFT ECE | SHIFT MCE | OOD AUROC | OOD FPR95 |
|---|---|---|---|---|
| V-TST, $l = 4$, $m = 10$ | 11.86±0.39 | 28.16±0.99 | 0.802±0.007 | 0.695±0.003 |
| V-TST, $l = 4$, $m = 1$ | 12.47±0.29 | **17.07±0.47** | 0.71±0.004 | 0.773±0.004 |
| TST, $l = 4$ | 13.45±0.04 | 30.25±0.09 | 0.875±0.002 | 0.727±0.013 |
| TST, $l = 4$, $Z = 10$ | 11.69±0.43 | 24.54±0.90 | 0.883±0.001 | 0.688±0.003 |
| TST, $l = 1$ | **11.46±0.27** | 25.35±0.65 | **0.884±0.001** | **0.682±0.002** |

## 5.4 Ablation Study: V-TST Dependence on Number of Samples in Training

To test whether using additional samples $n$ during V-TST improves the model calibration, we re-trained the base WRN-28-10 for CIFAR10 using V-TST with $n = 5$ MC samples and compared it to the original setup with $n = 1$. As shown in Table 6, using $n = 5$ results in slightly better MCE and OOD metrics, while the overall calibration performance remains comparable or slightly worse. Again, the Brier score was consistent with overall performance and aligned with the ECE and MCE calibration metrics. In contrast to our expectations, increasing the number of samples drawn during training did not prove to be beneficial for improving the calibration. However, we suspect that further exploration of a broader range of $n$ values would provide deeper insights to draw better conclusions. As expected, we also observed reduced variability in most evaluation metrics with $n = 5$.

Table 6: Effect of number of samples used in the second stage of training with V-TST. Evaluation metrics on in-distribution, shifted data, and OOD data. The value $n$ denotes the number of MC samples used during training, while $m$ represents the number of samples used during testing. We used $Z = 128$ as latent dimension.

| Model | Accuracy | ECE | MCE | Brier | SHIFT ECE | SHIFT MCE | OOD AUROC | OOD FPR95 |
|---|---|---|---|---|---|---|---|---|
| V-TST, $n = 5$, $m = 10$ | **92.45±0.02** | **1.53±0.081** | **13.90±1.979** | **0.0114±0.00002** | **9.65±0.21** | 20.55±0.48 | **0.852±0.002** | **0.697±0.002** |
| V-TST, $n = 5$, $m = 1$ | 91.04±0.06 | 1.62±0.103 | 18.89±6.873 | 0.0143±0.00009 | 12.57±0.22 | **20.08±0.39** | 0.762±0.004 | 0.779±0.004 |
| V-TST, $n = 1$, $m = 10$ | 92.44±0.02 | 1.38±0.100 | 21.26±6.44 | 0.0114±0.00004 | 9.41±0.3 | 20.33±0.66 | 0.851±0.003 | 0.702±0.002 |
| V-TST, $n = 1$, $m = 1$ | 90.9±0.03 | 1.78±0.148 | 24.80±8.981 | 0.0145±0.00007 | 12.53±0.29 | **20.19±0.56** | 0.766±0.003 | 0.778±0.003 |

## 5.5 Ablation Study: Focal Loss

We investigated how two-stage training compares to the focal loss, and whether combining the two implicit calibration methods could lead to improved performance. To test this, we experimented with using three configurations: using focal loss in both training stages, the first training stage only or the second training stage only. For better evaluation, we experimented with two different variants of focal loss: using a constant value for $\gamma$ (FL) and using an adaptive, sample-dependent schedule (FLA) as described in Mukhoti et al. (2020). In accordance with Mukhoti et al. (2020), we used $\gamma = 3$ for the constant loss and a sample-dependent scheme that uses $\gamma = 3$ for $\hat{p}_y \in [0.2, 1]$ and $\gamma = 5$ $\hat{p}_y \in [0, 0.2)$, where $\hat{p}_y$ denotes the probability of the network to predict the correct label $y$.

**Focal loss base models** First, we observed that a base model trained on constant focal loss (FL-B) or adaptive focal loss (FLA-B) is more calibrated on in-distribution and shifted data than a base model trained using cross-entropy loss (CE-B), demonstrating the expected calibration benefits of focal loss. However, V-TST with $m = 10$ and CE loss (V10-CE2) outperforms base models trained using focal loss (FL-B and FLA-B) in terms of both ECE and MCE scores as well as accuracy. The Brier scores of the focal loss base models were lower than the score of the cross-entropy loss base model, consistent with the overall calibration performance of these three models.

**Focal loss in two-stage training**  When combining the two methods we noticed two key observations. Firstly, using focal loss as training objective for both stages in two-stage training significantly worsens performance on several calibration metrics such as ECE, NLL, Brier score and (mostly) MCE. We noticed a similar trend when comparing two-stage focal loss models to the reproduction results obtained in Table 1. Hence, fully combining two-stage training with focal loss does generally not lead to improved calibration compared to a base model trained with focal loss or two-stage training using CE loss. However, secondly, we observed a significant improvement of calibration on shift and OOD data. As can be seen in Figure 2 the focal loss base models become overconfident after the second stage using focal loss, however, the plots suggest an overall tendency of models to be less confident on shift data. This is in contrast to other observations, where models become overconfident (Minderer et al., 2021) but explains the improved calibration of the overconfident model on the shift data.

**Focal loss base and cross-entropy in second-stage**  Using focal loss in the second stage on top of a focal loss base model worsens the calibration overall. Therefore, we tested whether combining a better-calibrated base model (compared to a base model trained with CE loss, see Table 7) with a second stage that uses CE loss would enhance calibration performance. We found that this combination (V10-FLA-CE, V1-FLA-CE, FLA-CE) generally improved both ECE and MCE significantly compared to the base model that used adaptive focal loss (FLA-B) and the corresponding CE baseline models (V10-CE2, V1-CE2, CE2) while maintaining a similar accuracy. For instance, including FLA in the first stage for TST (FLA-CE) improved ECE and MCE by 37.1% and 27.4%, respectively compared to the CE2 model (see Table 7).While the best ECE score was achieved for V-TST ($m = 10$), this was not consistent with the best test NLL, Brier score, and OOD performance. Results for using FL-B as a base model and performing second stage training with CE were similar to those using FLA-B as base model and are thus reported in the Appendix D.

Table 7: Results for including Focal Loss (FL) and Adaptive Focal Loss (FLA) in the training procedure. Test metrics and evaluation metrics on in-distribution data. The first loss before + sign indicates the loss used for training the base model prior to TST and V-TST. We used a latent dimension of $Z = 128$. The asterisks (*) indicate the best value across the table.

| Model | Loss | Model Name | Accuracy | ECE | MCE | Train NLL | Test NLL | Brier |
|---|---|---|---|---|---|---|---|---|
| V-TST, $m = 10$ | 3 FL | V10-FL2 | **91.77±0.03** | 11.72±0.148 | 20.94±0.934 | 0.2016±0.0018 | 0.3694±0.0016 | 0.0143±0.00007 |
| V-TST, $m = 1$ | 3 FL | V1-FL2 | 87.71±0.11 | 11.08±0.482 | 24.35±1.850 | 0.3128±0.0069 | 0.5185±0.0054 | 0.0213±0.00023 |
| TST | 3 FL | FL2 | 91.75±0.03 | 5.90±0.384 | 29.24±8.359 | 0.1261±0.0043 | 0.2796±0.0034 | 0.0130±0.00011 |
| Temp. WRN | 3 FL | FL-TS | **91.78** | 1.89 | **10.25*** | 0.0880 | **0.2524** | **0.0123** |
| WRN | 3 FL | FL-B | **91.78** | **1.57** | 27.45 | **0.0688** | 0.2604 | **0.0123** |
| V-TST, $m = 10$ | 5-3 FLA + CE | V10-FLA-CE | 92.04±0.01 | **0.80±0.045*** | 20.33±1.981 | 0.0775±0.0004 | 0.2861±0.0008 | 0.0122±0.00078 |
| V-TST, $m = 1$ | 5-3 FLA + CE | V1-FLA-CE | 90.11±0.04 | 1.52±0.129 | 20.42±6.653 | 0.1351 ±0.0012 | 0.4101±0.0038 | 0.0158±0.00007 |
| TST | 5-3 FLA + CE | FLA-CE | 92.13±0.03 | 2.10±0.032 | **14.92±2.292** | **0.0595±0.0001** | **0.2529±0.0004** | **0.0119±0.00002** |
| V-TST, $m = 10$ | 5-3 FLA | V10-FLA2 | 92.02±0.03 | 11.82±0.151 | 23.31±1.625 | 0.1964±0.0016 | 0.3692±0.0012 | 0.0141±0.00006 |
| V-TST, $m = 1$ | 5-3 FLA | V1-FLA2 | 87.78±0.08 | 12.08±0.327 | 23.75±0.839 | 0.3181±0.0044 | 0.5259±0.0031 | 0.0170±0.00008 |
| TST | 5-3 FLA | FLA2 | 92.06±0.04 | 5.96±0.340 | 27.73±6.487 | 0.1199±0.0035 | 0.2735±0.0 030 | 0.0126±0.00009 |
| Temp. WRN | 5-3 FLA | FLA-TS | **92.14** | **1.42** | 81.25 | 0.077 | **0.242*** | **0.0119** |
| WRN | 5-3 FLA | FLA-B | **92.14** | 1.97 | **22.83** | **0.0602** | 0.2539 | 0.0120 |
| V-TST, $m = 10$ | CE + 3 FL | V10-CE-FL | 92.40±0.02 | 9.84±0.235 | 26.92±6.064 | 0.1659±0.0022 | 0.3316±0.0020 | 0.0127±0.00008 |
| V-TST, $m = 1$ | CE + 3 FL | V1-CE-FL | 89.26±0.08 | 10.65±0.403 | **25.35±2.224** | 0.2667±0.0037 | 0.4703±0.0034 | 0.0156±0.00006 |
| TST | CE + 3 FL | CE-FL | **92.54±0.02** | 3.21±0.255 | 27.50±4.491 | **0.0822±0.0021** | **0.2483±0.0024** | **0.0112±0.00236*** |
| V-TST, $m = 10$ | CE | V10-CE2 | 92.44±0.02 | **1.38±0.100** | **21.26±6.44** | 0.0655±0.0004 | 0.2579±0.0017 | **0.0114±0.00004** |
| V-TST, $m = 1$ | CE | V1-CE2 | 90.9±0.03 | 1.78±0.148 | 24.80±8.981 | 0.1116±0.0012 | 0.3732±0.0041 | 0.0145±0.00007 |
| TST | CE | CE2 | **92.55±0.01*** | 3.34±0.016 | 24.68±3.883 | **0.0532±0.0002*** | 0.2516±0.0007 | 0.0115±0.00001 |
| Temp. WRN | CE | CE-TS | 92.52 | 4.7 | 25.2 | 0.0616 | 0.3024 | 0.0122 |
| WRN | CE | CE-B | 92.52 | 6.06 | 36.28 | 0.1031 | 0.5124 | 0.0133 |

**Cross-entropy loss base and focal loss in second-stage**  For completeness, we also tested how using constant focal loss in the second stage on top of a CE base model would perform (V10-CE-FL, V1-CE-FL, CE-FL) and found a similar behavior to using constant focal loss or adaptive focal loss in both stages of training (V10-FL2, V1-FL2, FL2, V10-FLA2, V1-FLA2, FLA2). Both ECE and MCE scores were higher than those obtained for CE-only two-stage training. An exception to this was the TST model, which reached a similar performance than the CE2 model. Here, Brier Score, NLL, OOD performance and ECE were aligned. Taken the results for these models together with the ones using focal loss in both stages together, we find the overall tendency that using focal loss in the second stage of training is not a good choice.

We hypothesize that the worsened calibration may be due to the way focal loss interacts with the fixed feature representations established in the first stage. Two-stage training relies on retraining the classification head to express confidence that aligns with the structure of the latent feature space. However, since focal loss penalizes all output logits to a low level (Tao et al., 2023b), it may be problematic when the feature extractor has produced well-separated clusters, as it prevents the classifier from confidently predicting clear cases. However, this should be explored and tested further.

Table 8: Evaluation metrics on shifted and OOD data for the effect of including Focal Loss (FL) and Adaptive Focal Loss (FLA) in the training procedure. The first loss before + sign indicates the loss used for training the base model prior to TST and V-TST. We used a latent dimension of $Z = 128$. The asterisks (*) indicate the best value across the table.

| Model | Loss | Model Name | Shift ECE | Shift MCE | OOD AUROC | OOD FPR95 |
|---|---|---|---|---|---|---|
| V-TST, $m = 10$ | 3 FL | V10-FL2 | 3.92±0.39 | 6.07±0.42 | 0.799±0.004 | 0.768±0.003 |
| V-TST, $m = 1$ | 3 FL | V1-FL2 | 2.96±0.15 | 4.97±0.31 | 0.701±0.002 | 0.850±0.003 |
| TST | 3 FL | FL2 | **2.44±0.10** | **4.75±0.29** | **0.876±0.001** | **0.672±0.004** |
| Temp. WRN | 3 FL | FL-TS | 6.28 | 11.89 | **0.875** | 0.701 |
| WRN | 3 FL | FL-B | 11.93 | 22.21 | 0.866 | 0.740 |
| V-TST, $m = 10$ | 5-3 FLA + CE | V10-FLA-CE | **10.01±0.16** | 19.66±0.37 | 0.825±0.002 | 0.727±0.003 |
| V-TST, $m = 1$ | 5-3 FLA + CE | V1-FLA-CE | 12.65±0.32 | **19.19±0.56** | 0.762±0.002 | 0.790±0.003 |
| TST | 5-3 FLA + CE | FLA-CE | 11.75±0.07 | 23.36±0.16 | **0.877±0.001** | **0.676±0.002** |
| V-TST, $m = 10$ | 5-3 FLA | V10-FLA2 | 4.13±0.36 | 6.79±0.42 | 0.828±0.004 | 0.714±0.004 |
| V-TST, $m = 1$ | 5-3 FLA | V1-FLA2 | **2.50±0.15** | **4.28±0.29** | 0.735±0.003 | 0.827±0.005 |
| TST | 5-3 FLA | FLA2 | 2.75±0.06 | 5.53±0.22 | **0.887±0.001*** | **0.625±0.004*** |
| Temp. WRN | 5-3 FLA | FLA-TS | 7.06 | 13.36 | 0.88 | 0.65 |
| WRN | 5-3 FLA | FLA-B | 12.74 | 24.30 | 0.88 | 0.69 |
| V-TST, $m = 10$ | CE + 3 FL | V10-CE-FL | 3.92±0.39 | 6.07±0.42 | 0.799±0.004 | 0.768±0.003 |
| V-TST, $m = 1$ | CE + 3 FL | V1-CE-FL | **2.25±0.27*** | **4.20±0.51*** | 0.745±0.004 | 0.810±0.005 |
| TST | CE + 3 FL | CE-FL | 2.44±0.10 | 4.75±0.29 | **0.876±0.001** | **0.672±0.004** |
| V-TST, $m = 10$ | CE | V10-CE2 | **9.41±0.3** | 20.33±0.66 | 0.851±0.003 | 0.702±0.002 |
| V-TST, $m = 1$ | CE | V1-CE2 | 12.53±0.29 | **20.19±0.56** | 0.766±0.003 | 0.778±0.003 |
| TST | CE | CE2 | 13.09±0.04 | 29.32±0.10 | 0.883±0.001 | **0.688±0.003** |
| Temp. WRN | CE | CE-TS | 16.44 | 37.93 | **0.885** | 0.698 |
| WRN | CE | CE-B | 20.06 | 45.62 | 0.874 | 0.708 |

**Focal loss performance on shifted data**  Incorporating focal loss in two-stage training in any form leads to significantly lower ECE and MCE for shifted data compared to focal loss base models and two-stage CE models (see Table 8). The only exceptions to this are the V-TST models that used adaptive focal loss in the first stage and CE loss in the second stage (V10-FLA-CE, V1-FLA-CE) but even these perform similarly well compared to V-TST models that only use CE loss (V10-CE2, V1-CE2). Another interesting observation was that models trained in a two-stage fashion with focal loss in both or one of the stages perform better on shifted data than in-distribution data. The exception for this trend were models trained with adaptive focal loss in the first stage and CE loss in the second stage (V10-FLA-CE, V1-FLA-CE, FLA-CE). This is a rather surprising finding as the WRN-28-10 trained with CE loss (V10-CE2, V1-CE2, CE2) was better calibrated for in-distribution than shifted data for both CIFAR10 and CIFAR100 (see Table 1).

A possible explanation for the better performance on shifted data is that focal loss emphasizes hard examples during training (Lin et al., 2017), leading to better calibration on challenging shifted data but poorer calibration on easier in-distribution data. Additionally, focal loss with a fixed $\gamma$ has different dynamics than the CE loss. Initially, models trained with focal loss show higher weight norms than those trained with CE loss. However, this trend reverses after further training so that the weight norms of the focal loss model become lower, which can be explained by the regularizing effect the focal loss has on the network weights (Mukhoti et al., 2020).

**Focal loss performance on OOD data**  For the OOD data, we find two interesting results. First, TST generally performs best both in terms of AUROC and FPR95, regardless of the loss function combination

(even the CE baseline model (CE-B) has the lowest FPR95 score for the TST). This shows the superiority of TST on OOD data. In contrast, V-TST ($m = 1$) performs worst in all configurations for both AUROC and FPR95. Second, among the TST models, using adaptive focal loss in both stages of training (FLA2) reaches the highest AUROC and lowest FPR95 rate overall, outperforming both the CE baseline and the other focal loss combinations. Thirdly, the combination of adaptive focal loss in the first stage and CE loss in the second stage (V10-FLA-CE, V1-FLA-CE, FLA-CE) mostly worsens the performance (both AUROC and FPR95) compared to using the adaptive focal loss in the first and second stages of training (V10-FLA2, V1-FLA2, FLA2) or the CE baseline (CE-B).

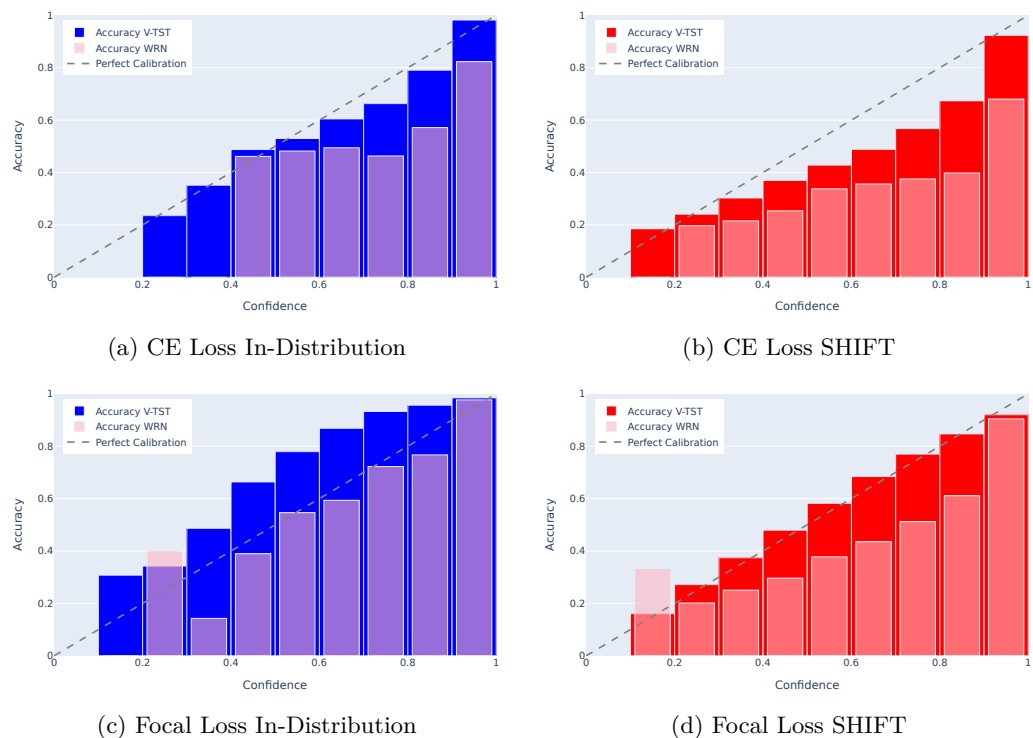

(a) CE Loss In-Distribution         (b) CE Loss SHIFT

(c) Focal Loss In-Distribution         (d) Focal Loss SHIFT

Figure 2: ECE plots for models with CE and focal loss. (a) In-distribution ECE plot of a V-TST trained model ($Z = 128$, $m = 10$, V10-CE2) and the base WRN-28-10 trained on CIFAR10 with CE loss (CE-B). (b) Same models with CE loss evaluated on shifted data CIFAR10-C. (c) In-distribution ECE plot of a V-TST trained model ($Z = 32$, $m = 1$, V1-FLA-B) and the base WRN-28-10 trained on CIFAR10 with focal loss (FLA-B). (d) Same models with focal loss evaluated on shifted data CIFAR10-C.

**Reliability diagrams** Figure 2 shows the ECE plots for the base and V-TST models trained with CE loss and the 5-3 focal loss in both stages. Starting from an uncalibrated WRN-28-10 model trained with CE loss in 2a (CE-B), V-TST in the second stage leads to an improved calibration (V10-CE2). 2c shows that the base WRN-28-10 that was trained with adaptive focal loss (FLA-B) is already better calibrated than the base WRN-28-10 model trained with CE (CE-B) in 2a. The second training stage with adaptive focal loss makes the model underconfident in its predictions but also evens out the overconfidence it showed for images that were around 35% certain. For adaptive focal loss on the shifted dataset (2d), both the base WRN-28-10 (FLA-B) and V-TST (V1-FLA-B) models perform worse than on in-distribution data, with a decrease in accuracy. However, for the V-TST model (V1-FLA-B), this results in its confidence aligning more closely with its accuracy, whereas the base model becomes overconfident.

## 5.6 Ablation Study: Label Smoothing

Following the results from combining focal loss and two-stage training, we investigated whether label smoothing, another implicit regularization technique, can be similarly combined with two-stage training to improve

model calibration. In order to verify this, we experimented with two configurations: using label smoothing in both training stages, and only in the second stage. We chose a smoothing factor $\alpha = 0.1$, consistent with prior work on similar network architectures (Müller et al., 2019).

Table 9: Results for including Label Smoothing (LS) in the training procedure. Test metrics and evaluation metrics on in-distribution data. The method column specifies the training strategy. If a single entry is present, it indicates that the loss function or regularization is applied consistently across both training stages. If two entries are separated by a +, the first denotes the configuration for the first stage and the second for the second stage. We used a latent dimension of $Z = 128$. The asterisks (*) indicate the best value across the table.

| Model | Method | Model Name | Accuracy | ECE | MCE | Train NLL | Test NLL | Brier |
|---|---|---|---|---|---|---|---|---|
| V-TST, $m=10$ | LS | V10-LS2 | 92.49±0.01 | 9.00±0.100 | 24.88±6.254 | 0.1938±0.0008 | 0.3797±0.0009 | 0.0132±0.00002 |
| V-TST, $m=1$ | LS | V1-LS2 | 92.37±0.02 | 8.26±0.134 | 26.21±5.897 | 0.2054±0.0011 | 0.4134±0.0020 | 0.0146±0.00008 |
| TST | LS | LS2 | **92.50±0.00** | **6.01±0.176** | **19.57±0.767** | **0.1358±0.0010** | **0.3321±0.0013** | **0.0121±0.00003** |
| Temp. WRN | LS | LS-TS | 92.22 | 9.04 | 26.93 | 0.1891 | 0.3796 | 0.0133 |
| WRN | LS | LS-B | 92.22 | 6.71 | 27.09 | 0.1446 | 0.3523 | 0.0129 |
| V-TST, $m=10$ | CE + LS | V10-CE-LS | 92.17±0.03 | 6.98±0.155 | 21.18±2.947 | 0.1503±0.0018 | 0.3425±0.0016 | 0.0127±0.00003 |
| V-TST, $m=1$ | CE + LS | V1-CE-LS | 91.61±0.05 | 5.89±0.170 | 24.64±4.196 | 0.1625±0.0019 | 0.3836±0.0015 | 0.0143±0.00004 |
| TST | CE + LS | CE-LS | **92.21±0.02** | **3.42±0.201** | **14.60±1.890*** | **0.1046±0.0030** | **0.2919±0.0043** | **0.0119±0.00004** |
| V-TST, $m=10$ | CE | V10-CE2 | 92.44±0.02 | **1.38±0.100*** | **21.26±6.44** | 0.0655±0.0004 | 0.2579±0.0017 | **0.0114±0.00004*** |
| V-TST, $m=1$ | CE | V1-CE2 | 90.9±0.03 | 1.78±0.148 | 24.80±8.981 | 0.1116±0.0012 | 0.3732±0.0041 | 0.0145±0.00007 |
| TST | CE | CE2 | **92.55±0.01*** | 3.34±0.016 | 24.68±3.883 | **0.0532±0.0002*** | **0.2516±0.0007*** | 0.0115±0.00001 |
| Temp. WRN | CE | CE-TS | 92.52 | 4.7 | 25.2 | 0.0616 | 0.3024 | 0.0122 |
| WRN | CE | CE-B | 92.52 | 6.06 | 36.28 | 0.1031 | 0.5124 | 0.0133 |

Table 10: Evaluation metrics on shifted and OOD data for the effect of including Label Smoothing (LS) in the training procedure. The method column specifies the training strategy. If a single entry is present, it indicates that the loss function or regularization is applied consistently across both training stages. If two entries are separated by a +, the first denotes the configuration for the first stage and the second for the second stage. We used a latent dimension of $Z = 128$. The asterisks (*) indicate the best value across the table.

| Model | Method | Model Name | Shift ECE | Shift MCE | OOD AUROC | OOD FPR95 |
|---|---|---|---|---|---|---|
| V-TST, $m=10$ | LS | V10-LS2 | **5.04±0.15** | 21.52±0.32 | 0.719±0.009 | 0.752±0.002 |
| V-TST, $m=1$ | LS | V1-LS2 | 5.64±0.24 | **8.65±0.45** | 0.646±0.006 | 0.811±0.004 |
| TST | LS | LS2 | 10.67±0.16 | 26.64±0.47 | **0.831±0.005** | **0.748±0.001** |
| Temp. WRN | LS | LS-TS | 6.91 | 31.62 | 0.611 | 0.763 |
| WRN | LS | LS-B | 11.06 | 34.93 | 0.612 | 0.763 |
| V-TST, $m=10$ | CE + LS | V10-CE-LS | **3.12±0.07*** | 11.10±0.22 | 0.821±0.003 | 0.688±0.003 |
| V-TST, $m=1$ | CE + LS | V1-CE-LS | 4.19±0.11 | **5.88±0.28*** | 0.767±0.003 | 0.727±0.003 |
| TST | CE + LS | CE-LS | 5.49±0.27 | 16.19±0.98 | **0.863±0.006** | **0.657±0.005*** |
| V-TST, $m=10$ | CE | V10-CE2 | **9.41±0.3** | 20.33±0.66 | 0.851±0.003 | 0.702±0.002 |
| V-TST, $m=1$ | CE | V1-CE2 | 12.53±0.29 | **20.19±0.56** | 0.766±0.003 | 0.778±0.003 |
| TST | CE | CE2 | 13.09±0.04 | 29.32±0.10 | 0.883±0.001 | **0.688±0.003** |
| Temp. WRN | CE | CE-TS | 16.44 | 37.93 | **0.885*** | 0.698 |
| WRN | CE | CE-B | 20.06 | 45.62 | 0.874 | 0.708 |

The results in Table 9 show that using LS alone slightly improves the calibration of the base WRN model on in-distribution data, particularly in terms of MCE, Test NLL, and Brier score. However, when combined with TS (LS-TS) or V-TST (V1-LS2, V10-LS), the calibration performance of the base model (LS-B) decreases across nearly all metrics and is worse than the ones obtained using CE loss alone (CE-TS, V1-CE2, V10-CE2). While LS does benefit from TST (LS2), the improvements are still smaller compared to using TST with CE loss alone (CE2), except for MCE. Notably, including LS into the second stage of (V)-TST improves the MCE compared to the results obtained using CE loss alone (CE2, V10-CE2, V1-CE2), likely due to its tendency to reduce prediction overconfidence. For TST using LS only in the second stage (CE-LS), the

performance is comparable to using CE loss alone in both stages (CE2) across all metrics, with a significant improvement in MCE.

Under distribution shifts and OOD data, using LS in the second stage leads to better overall performance as shown in Table 10. When evaluated on shift data, using LS largely improves the ECE and MCE metrics compared to the results obtained with CE loss alone. Similarly to the in-distribution results, using LS only in the second stage leads to better performance than using it in both stages. In both setups, the V-TST models (V10-LS2, V1-LS2, V10-CE-LS, V1-CE-LS) perform best, with $m = 1$ achieving the best MCE and $m = 10$ the best ECE scores. For OOD data, using LS in the second stage (V10-CE-LS, V1-CE-LS, CE-LS) improves the FPR95 metric compared to the CE results (V10-CE2, V1-CE2, CE2) but leads to a slight drop in AUROC, whereas using LS in both stages worsens both metrics.

In summary, we found that integrating LS in two-stage training does not lead to further improvements in calibration on in-distribution data compared to the reproduced results. The exception is applying LS only in the second stage of TST. Under distribution shifts and OOD data, using LS in the second stage leads to better overall performance.

## 5.7 Ablation Study: L2 regularization

Lastly, we investigated whether a similar pattern can be found for combining two-stage training with an explicit regularization technique such as L2 regularization. Again, we experimented with using L2 regularization during both stages of training or only in the second stage of training. We chose $\lambda = 0.003$, based on the results of Joo & Chung (2020) who performed a hyperparameter search for ResNet-50 on CIFAR10 to find the best L2 regularization strength.

The results show that the use of L2 regularization leads to a better calibrated base model (L2-B) for the in-distribution data in terms of ECE and MCE scores, but increased Brier score (see Table 11). For the OOD data, the base model that used L2 regularization performs worse compared to the CE-B model. No clear trend can be seen for the base models in regards to shift data.

Table 11: Results for including L2 regularization (L2) in the training procedure. Test metrics and evaluation metrics on in-distribution data. The method column specifies the training strategy. If a single entry is present, it indicates that the loss function or regularization is applied consistently across both training stages. If two entries are separated by a +, the first denotes the configuration for the first stage and the second for the second stage. We used a latent dimension of $Z = 128$. The asterisks (*) indicate the best value across the table.

| Model | Method | Model Name | Accuracy | ECE | MCE | Train NLL | Test NLL | Brier |
|---|---|---|---|---|---|---|---|---|
| V-TST, $m = 10$ | L2 | V10-L2-2 | **90.67±0.02** | **2.58±0.171** | **15.90±0.591** | 0.1127±0.0010 | **0.3578±0.0011** | **0.0146±0.00002** |
| V-TST, $m = 1$ | L2 | V1-L2-2 | 90.47±0.05 | 3.53±0.222 | 18.29±1.873 | 0.1305±0.0003 | 0.4232±0.0046 | 0.0157±0.00009 |
| TST | L2 | L2-2 | 90.66±0.01 | 4.32±0.153 | 28.79±5.478 | **0.1052±0.0037** | 0.3590±0.0011 | 0.0148±0.00005 |
| Temp. WRN | L2 | L2-TS | 90.16 | 2.24 | 19.27 | 0.1531 | 0.3678 | 0.0152 |
| WRN | L2 | L2-B | 90.16 | 5.00 | 27.37 | 0.1160 | 0.3865 | 0.0159 |
| V-TST, $m = 10$ | CE + L2 | V10-CE-L2 | 92.09±0.01 | **1.04±0.106*** | **8.79±1.374*** | **0.0660±0.0005** | 0.2647±0.0010 | **0.0118±0.00002** |
| V-TST, $m = 1$ | CE + L2 | V1-CE-L2 | 91.21±0.04 | 3.11±0.160 | 14.98±1.578 | 0.0822±0.0005 | 0.3409±0.0043 | 0.0137±0.00008 |
| TST | CE + L2 | CE-L2 | **92.26±0.02** | 2.60±0.220 | 19.62±3.328 | 0.0567±0.0003 | **0.2573±0.0024** | 0.0119±0.00004 |
| V-TST, $m = 10$ | CE | V10-CE2 | 92.44±0.02 | 1.38±0.100 | **21.26±6.44** | 0.0655±0.0004 | 0.2579±0.0017 | **0.0114±0.00004*** |
| V-TST, $m = 1$ | CE | V1-CE2 | 90.9±0.03 | 1.78±0.148 | 24.80±8.981 | 0.1116±0.0012 | 0.3732±0.0041 | 0.0145±0.00007 |
| TST | CE | CE2 | **92.55±0.01*** | 3.34±0.016 | 24.68±3.883 | **0.0532±0.0002*** | **0.2516±0.0007*** | 0.0115±0.00001 |
| Temp. WRN | CE | CE-TS | 92.52 | 4.7 | 25.2 | 0.0616 | 0.3024 | 0.0122 |
| WRN | CE | CE-B | 92.52 | 6.06 | 36.28 | 0.1031 | 0.5124 | 0.0133 |

Applying L2 regularization in both stages results in worse calibration on in-distribution data for TST (L2-2), compared to the CE baseline (CE2). Similarly, for V-TST (V10-L2-2, V1-L2-2), using L2 regularization worsens ECE, NLL, and Brier scores compared to not using L2 regularization (V10-CE-2, V1-CE-2, CE-2, respectively). However, MCE scores improve for these V-TST models.

Using L2 in the second stage of training on top of the CE base model leads to improved calibration in terms of ECE and MCE scores compared to using L2 in both stages and not using L2 at all (see Table 11). The

only exception is the V1-CE2 model, which produces a lower ECE score compared to the V1-CE-L2 model. Moreover, using L2 regularization only in the second stage also improves the Brier score compared to using L2 in both stages. Hence, using L2 regularization in the second stage tends to improve calibration, which is in contrast to the results obtained using focal loss.

Under distribution shift, using L2 regularization in both stages worsens ECE and MCE compared to the results obtained without the explicit regularization, as can be seen in Table 12. However, using L2 regularization in the second stage only mostly improves ECE and MCE except for V-TST with $m = 1$, where it increases both (V1-CE-L2 compared to V1-CE2). The worst OOD detection performance is associated with using L2 regularization in both stages, while the best one is achieved by using it only in the second stage.

Taken together, applying L2 regularization in the second stage can considerably improve calibration compared to using it in both or none of the stages. These results extend from the in-distribution data to shift data and OOD detection. Interestingly, this finding is in contrast to the results of the focal loss and, to a lesser extent, LS ablation studies, where using the implicit regularization technique tends to worsen the in-distribution calibration of V-TST compared to the CE baseline if it is applied in the second stage of training.

Table 12: Evaluation metrics on shifted and OOD data for the effect of including L2 regularization (L2) in the training procedure. The method column specifies the training strategy. If a single entry is present, it indicates that the loss function or regularization is applied consistently across both training stages. If two entries are separated by a +, the first denotes the configuration for the first stage and the second for the second stage. We used a latent dimension of $Z = 128$. The asterisks (*) indicate the best value across the table.

| Model | Method | Model Name | Shift ECE | Shift MCE | OOD AUROC | OOD FPR95 |
|---|---|---|---|---|---|---|
| V-TST, $m = 10$ | L2 | V10-L2-2 | 18.30±0.21 | 34.56±0.36 | 0.743±0.002 | 0.765±0.001 |
| V-TST, $m = 1$ | L2 | V1-L2-2 | 20.50±0.31 | 33.39±0.61 | 0.713±0.001 | 0.780±0.002 |
| TST | L2 | L2-2 | 20.54±0.67 | 38.37±1.18 | 0.764±0.002 | 0.762±0.002 |
| Temp. WRN | L2 | L2-TS | **14.68** | **26.23** | **0.825** | **0.725** |
| WRN | L2 | L2-B | 23.02 | 40.45 | 0.824 | 0.741 |
| V-TST, $m = 10$ | CE + L2 | V10-CE-L2 | **8.37±0.25*** | 17.21±0.42 | 0.887±0.001 | 0.637±0.002 |
| V-TST, $m = 1$ | CE + L2 | V1-CE-L2 | 12.92±0.33 | **25.42±0.67** | 0.862±0.001 | 0.687±0.003 |
| TST | CE + L2 | CE-L2 | 11.80±0.49 | 25.72±1.19 | **0.897±0.001*** | **0.632±0.002*** |
| V-TST, $m = 10$ | CE | V10-CE2 | **9.41±0.3** | 20.33±0.66 | 0.851±0.003 | 0.702±0.002 |
| V-TST, $m = 1$ | CE | V1-CE2 | 12.53±0.29 | **20.19±0.56*** | 0.766±0.003 | 0.778±0.003 |
| TST | CE | CE2 | 13.09±0.04 | 29.32±0.10 | 0.883±0.001 | **0.688±0.003** |
| Temp. WRN | CE | CE-TS | 16.44 | 37.93 | **0.885** | 0.698 |
| WRN | CE | CE-B | 20.06 | 45.62 | 0.874 | 0.708 |

## 6 Conclusion

In this paper, we aimed to reproduce the main results from Jordahn & Olmos (2024) and extended their findings by performing numerous ablation studies to get new insights into two-stage training. We confirmed their findings that decoupling the training of feature and classification layers improves the calibration of a WRN-28-10 model on CIFAR10 and CIFAR100, with slightly varying effects on CIFAR100. Additionally, we successfully reproduced two of their ablation studies, the CNN ablation study and the ViT fine-tuning ablation study. From the additional ablation studies in this paper, we found that the effects of improved calibration from two-stage training transfer to other similar model architectures, namely ResNet-50 and EfficientNet-B5. However, the effectiveness of two-stage training differs across CNN architectures. For the second stage architecture, we observed the importance of balancing the number of layers in the MLP with the latent dimension $Z$ used. Lastly, we found that using more MC samples during training does not affect the calibration performance of the models positively. Thus, overall, the performance was less affected by our ablation studies than the choice of $Z$, as could be seen from their additional results.

Finally, we followed the suggestion of Jordahn & Olmos (2024) to combine their method with other calibration techniques. Specifically, we explored the integration of their approach with two implicit regularization

methods, focal loss and label smoothing, and one explicit regularization method, L2 regularization. Our investigation led to four key takeaways in regards to in-distribution data. First, all regularization methods improve the calibration of the base WRN-28-10. Second, using additional implicit regularization in both stages of training tends to perform worse than the CE baseline or the corresponding base model before the second stage. Third, combining two-stage training with additional regularization can either enhance or degrade calibration, depending on the regularization and in which stages it is integrated. For instance, focal loss, when used only in the second stage of V-TST, worsens calibration of the corresponding base models, whereas L2 regularization tends to have the opposite effect. Lastly, applying focal loss only to the base model, followed by TST or V-TST using CE loss, resulted in better-calibrated models compared to the baseline. When looking at the OOD detection performance, we found a tendency for TST to perform better in OOD detection than V-TST and the corresponding base model across most experiments. On the other hand, V-TST generally achieved better performance under distribution shifts. Notably, TST also balances in-distribution and shift performance well when additional implicit regularization is used in the second stage of training.

To conclude, we found that TST and V-TST can further improve calibration when combined with another regularization method. However, both the choice of regularization method and the way it is combined with TST or V-TST significantly influence the method's effectiveness on calibration. We had to conduct several experiments to identify a good combination, and our initial attempts even worsened calibration. Nonetheless, the approach remains promising, and our ablation studies extend the empirical results of Jordahn & Olmos (2024). A thorough theoretical analysis of these methods remains open for future work.

## 7 Future Work

Next to the theoretical analysis of (V-)TST, future extensions should also investigate possible explanations as to why using focal loss for both stages of training decreased model calibration compared to the focal loss base model. One possible next step could be to apply early stopping based on ECE (as has been tested by Mukhoti et al. (2020)) during the second stage of training to avoid an over-regularization of the weights. Other ideas would be testing whether the obtained focal loss results are transferable to other model architectures beyond WRN-28-10, or further experimenting with the second stage latent dimension size and other datasets.

Additionally, we would be interested to see further work investigating how the TST and V-TST perform when combined with other implicit regularization techniques such as OKO (Muttenthaler et al., 2023), or post-hoc calibration methods such as laplace approximation, which had also been suggested in Jordahn & Olmos (2024). They claimed that their method is not an alternative to other techniques but rather complementary to them. However, our results show that combining TST and V-TST with other calibration techniques is not always straightforward but requires careful experimental design.

## 8 Challenges and Limitations

During the study, we encountered some challenges in replicating the results from Jordahn & Olmos (2024) due to missing information and incomplete code, which required us to re-implement parts of it and make assumptions about the training hyperparameters. For instance, details on the MLP head design for the CNN in TST and V-TST, choices for latent dimension, training seeds for the base WRNs, and information regarding the pre-trained ViT model for fine-tuning were omitted. Additionally, the model structures reported in Tables 1 and 2 were not immediately apparent. The variations in the latent dimension $Z$ for each model could only be seen by inspecting and comparing all results in the Appendix in detail.

The theoretical justification for the proposed methods by Jordahn & Olmos (2024) could have been discussed in more detail. While the authors provide some intuition for why the decoupled learning of feature and classification layers improves calibration, they offer limited reasoning for selecting the latent dimension. Although they conduct multiple experiments with different $Z$, the reasoning behind the choice of the latent dimension is unclear, despite different values of $Z$ leading to significant variation in the results.

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

## A  Datasets

Table 13: Overview of used datasets and corresponding preprocessing steps.

| Dataset | # Images | # Classes | Size | Preprocessing | Implementation |
|---|---|---|---|---|---|
| CIFAR10 | 60,000 | 10 | 32x32 | Normalization, Random Crop, Random Horizontal Flip | PyTorch |
| CIFAR100 | 60,000 | 100 | 32x32 | Normalization, Random Crop, Random Horizontal Flip | PyTorch |
| SVHN | 600,000 | 10 | 32x32 | Normalization | PyTorch |
| CIFAR10-C | 60,000 | 10 | 32x32 | Corruption Added | Downloaded |
| CIFAR100-C | 60,000 | 100 | 32x32 | Corruption Added | Downloaded |
| Tiny ImageNet | 100,000 | 200 | 64x64 | Resize (224x224), Normalization | GitHub Repo |

## B  Reproducing results for CIFAR100

We noticed that our WRN-28-10 base models for CIFAR100 achieved the lowest validation loss in the early epochs (14, 16, 17, 18, 18, 20, 28 and 61) even though the training accuracy reached around 100% and the validation accuracy kept improving to similar levels reported in the original paper. The model reported in Table 1 is the one that achieved the lowest validation loss in epoch 61 (seed 2). In Table 14 and Table 15 we report the results when using seed 1 (our default seed for all other models). As we tried various seeds, used the code provided by the authors, and followed their instructions, we were surprised to see that our base models never reached their reported accuracy. Similar results may have been obtainable if we had known their seed or if we had saved the model with the highest validation accuracy rather than the model with the lowest validation loss. However, we did not explore the latter possibility further, as the original paper also saved the models based on the lowest validation accuracy, and because this could have had negative effects on calibration.

Table 14: Replication of CIFAR100 results with **seed 1**. Test metrics on in-distribution test datasets. For V-TST, $m = 1$ and $m = 10$ indicate using 1 and 10 samples in the MC approximation, respectively. Temp. WRN is the temperature-scaled WRN. $Z = 512$ was used for V-TST ($m = 10$) and TST, and $Z = 128$ for V-TST ($m = 1$).

| Dataset | Model | Accuracy | ECE | MCE | Train NLL | Test NLL | Brier |
|---|---|---|---|---|---|---|---|
| | V-TST, $m = 10$ | **64.32**±0.09 | 9.92±0.288 | 21.74±1.349 | **0.5008**±**0.0082** | 1.4159±0.0034 | **0.0049**±**0.0034** |
| | V-TST, $m = 1$ | 62.35±0.07 | 10.71±0.318 | 20.48±0.780 | 0.6064±0.0049 | 1.5634±0.0045 | 0.0052±0.00003 |
| CIFAR100 | TST | 61.82±0.16 | 10.32±0.560 | 20.55±2.404 | 0.6448±0.0098 | 1.5212±0.0067 | 0.0052±0.0000008 |
| | Temp. WRN | 52.31 | **2.72** | **6.42** | 1.2029 | 1.8358 | 0.0062 |
| | WRN | 52.31 | 17.38 | 27.52 | 1.1451 | 2.0083 | 0.0066 |

Due to the accuracy differences between our CIFAR100 model and the original one, the typical calibration measures ECE and MCE cannot be directly applied to compare their calibration. This is because ECE is an estimator whose bias is dependent on the accuracy and is highest for models with an accuracy around

Table 15: Replication of CIFAR100 results with **seed 1**. Evaluation metrics on shifted data and OOD data.

| Dataset | Model | SHIFT ECE | SHIFT MCE | OOD AUROC | OOD FPR95 |
|---------|-------|-----------|-----------|-----------|-----------|
| | V-TST, $m = 10$ | 20.30±0.42 | 32.39±0.66 | **0.727±0.002** | **0.817±0.002** |
| | V-TST, $m = 1$ | 21.15±0.40 | 34.01±0.60 | 0.717±0.002 | 0.828±0.002 |
| CIFAR100 | TST | 21.13±0.76 | 30.82±1.13 | 0.692±0.002 | 0.846±0.004 |
| | Temp. WRN | **11.09** | **20.46** | 0.677 | 0.878 |
| | WRN | 28.15 | 43.18 | 0.658 | 0.885 |

50%, while it decreases for models with more extreme accuracies (closer to 0 or 1) (Minderer et al., 2021). Therefore, one should always consider both accuracy and the calibration measure when comparing models with different classification errors and choose the bin size carefully (Minderer et al., 2021). Here, we use the same bin size ($M = 10$) as in the original paper for comparability. Alternative calibration measures such as the NLL or the Brier score don't suffer from the same bias and should be taken into account as well (Minderer et al., 2021).

## C  Network architecture details

**ViT architecture**  As the original study does not specify which weights or latent dimension they used, we decided to use the weights of google/vit-base-patch16-224-in21k from the transformers package which corresponds to the ViT-B/16 architecture and used the latent dimension $Z = 128$ for both V-TST and TST. As Tiny ImageNet images are too small for the ViT, we had to resize them before using them as an input to the model. We used the ViTForImageClassification and the ViTModel class from the transformer package to include the ViT in the code base of the original paper.

**ResNet-50 and EfficientNet-B5 architecture**  The architectures used for ResNet-50 and EfficientNet-B5 have been adapted to the smaller input image size of the CIFAR10 dataset. For ResNet-50, the parameters of the first convolutional layers have parameters: stride 1, padding 1, kernel size of $3 \times 3$. No max-pooling is performed after the first convolution and global average pooling is performed over a smaller feature map of $4 \times 4$.

For EfficientNet-B5, the first convolutional layer has a stride and padding of 1 and uses a kernel size of $3 \times 3$. Similarly, the first down-sampling layer following the first convolutional layer is removed and the feature map of the global average pooling is reduced to $2 \times 2$.

For both ResNet and EfficientNet the number of output classes is changed to 10.

**MLP Head Modifications**  For the single-layer configuration, we simply re-initialized the final FC layer. For the four-layer configuration, we extended the original three-layer MLP to have dimensions [`output`, $6Z$], [$6Z$, $3Z$], [$3Z$, $Z$] and [$Z$, #classes].

## D  Additional Results

Table 16: Test metrics on in-distribution data for the effect of including Focal Loss (FL) in the training procedure. The first loss before + sign indicates the loss used for training the base model prior to TST and V-TST.

| Dataset | Model Name | Model | Loss | Accuracy | ECE | MCE | Train NLL | Test NLL |
|---------|-----------|-------|------|----------|-----|-----|-----------|----------|
| CIFAR10 | V-TST, $m = 10$ | V10-FL-CE | 3 FL + CE | **91.88±0.02** | **0.77±0.054** | 20.60±2.488 | 0.0821±0.0006 | 0.2840±0.0011 |
| | V-TST, $m = 1$ | V1-FL-CE | 3 FL + CE | 89.82±0.07 | 1.52±0.076 | 25.68±8.816 | 0.1405±0.0017 | 0.4124±0.0023 |
| | TST | FL-CE | 3 FL + CE | 91.86±0.01 | 2.03±0.020 | **18.48±3.253** | **0.0652±0.0002** | **0.2577±0.0005** |

# E  Replication of Ablation Studies

Table 17: Replication of test metrics for ablation studies on in-distribution data test-sets using a simple CNN on CIFAR10. We used $Z = 512$ for V-TST with $m = 10$, $Z = 128$ for V-TST with $m = 1$, and $Z = 32$ for TST.

|  | Model | Accuracy | ECE | MCE |
|---|---|---|---|---|
| | V-TST, $m = 10$ | **71.25±0.04** | 1.82±0.07 | 8.85±1.42 |
| | V-TST, $m = 1$ | 67.85±0.07 | 1.25±0.08 | **6.55±1.17** |
| Ours | TST, $Z = 128$ | 71.03±0.04 | 1.11±0.05 | 18.64±0.49 |
| | TST | 70.55±0.05 | **0.73±0.05** | 14.80±1.98 |
| | CNN | 70.83 | 0.80 | 19.46 |
| | V-TST, $m = 10$ | **75.95±0.2** | 3.27±0.05 | **10.65±4.35** |
| | V-TST, $m = 1$ | 73.22±0.1 | 1.61±0.3 | **9.08±5.47** |
| Original | TST | 70.77±0.2 | **0.76±0.17** | **7.79±4.67** |
| | CNN | 72.48 | 1.04 | 19.89 |

Table 18: Replication of test metrics for ablation studies on in-distribution data test-sets while fine-tuning a ViT on Tiny ImageNet. We used $Z = 128$ for V-TST and TST.

|  | Model | Accuracy | ECE | MCE |
|---|---|---|---|---|
| | V-TST, $m = 10$ | **84.36±0.04** | 3.35±0.07 | 12.46±0.81 |
| | V-TST, $m = 1$ | 83.64±0.08 | **1.56±0.08** | **11.13±1.17** |
| Ours | TST | 83.98±0.05 | 6.11±0.11 | 22.03±2.03 |
| | FT ViT-B/16 | 83.38 | 3.33 | 13.60 |
| | V-TST, $m = 10$ | **86.7±0.02** | 1.96±0.07 | 9.76±0.56 |
| | V-TST, $m = 1$ | 85.63±0.05 | **2.3±0.27** | **8.62±0.55** |
| Original | TST | 85.99±0.07 | 3.47±0.44 | 14.77±1.69 |
| | FT ViT-B/16 | 85.46 | 2.19 | 12.54 |

