# OpenReview forum: "A Reproducibility Study of Decoupling Feature Extraction and Classification Layers for Calibrated Neural Networks"
_TMLR — Accepted by TMLR_

### Review · Reviewer_UyE4 · 2025-03-26

**Summary Of Contributions:**

This paper replicates the existing work (Jordahn & Olmos, 2024) results and proposes combining the focal loss with (V-)TST, which is the method by Jordahn & Olmos, to further improve overconfidence and calibration in image classification.
This paper also extends the experiments by Jordahn & Olmos with more ablation and model variants.

**Audience:**

No

**Claims And Evidence:**

No

**Requested Changes:**

## Major changes
- For W1 and W2: Clarifying the motivation of replicating Jordahn & Olmos's results is necessary. It would also be compelling to explain why TST is essential and worth replicating since many calibration studies exist. If there was any problem when replicating TST, describing the problem and how it was addressed would support the contribution of the replication.
- For W3: Unveiling the problem of TST and describing how the focal loss solves the TST's problem would be interesting and strengthen the reasonability of the method. The current paper just combines the two methods and not motivated.
- For W4: Reviewing other existing calibration studies, explaining their problems, and explaining why TST and focal loss are focused on in this paper would strengthen the motivation.
- For W5: Experimenting with datasets not used in Jordahn & Olmos's paper or evaluating with other metrics such as Brier score would bring new insights. For instance, Cars, Pets, Food, Birds, Flowers, domain adaptation datasets, etc., would be beneficial for practitioners.


## Minor changes
- The citation of the most important work in this paper, Jordahn & Olmos (2024), is inaccurate. It has been [accepted to ICML2024](https://proceedings.mlr.press/v235/jordahn24a.html) but cited as an arxiv paper.
- Adding the scores from the original paper to Tabs. 1-4 would be helpful for checking the replication.

**Strengths And Weaknesses:**

## Strengths
- S1: Experimental settings are well described.

## Weaknesses
- W1: Replication of TST itself is not a contribution since Jordahn & Olmos released the code. Is there any problem when running the code or suspicion about the results?
- W2: The motivation for the replication is unclear. Clarifying the importance of (V-)TST and why we should replicate the results would strengthen the motivation.
- W3: The justification of the proposed method's design, combining the focal loss with TST, is unclear.
- W4: The related work section should be more extensive. Reviewing existing calibration studies and the importance of the TST and focal loss would strengthen the paper.
- W5: For examining TST, experimenting with datasets not used in Jordahn & Olmos's paper or evaluating with other metrics such as Brier score [a] would bring new insights.


[a] https://en.wikipedia.org/wiki/Brier_score

---

> ### Author Response · Authors · 2025-04-20
>
> We want to thank you for your feedback and time to review our submission. We have, to the best of our ability, tried to improve and change our submission in accordance to your feedback.
>
> These are the changes we made:
>
> - **W1 and W2:** We described our motivation in section 3.3 in more detail. To summarize, we think that (V-)TST is a promising technique as it outperformed another post-hoc method and can be applied to a broad range of models with little computational overhead. Further, the original paper is not based on theoretical derivations, and thus a replication of the empirical study can help to verify the validity of the obtained results.
>
> - **W3:** The motivation behind combining those two methods came from the suggestion of the original paper that their method should be seen as  complementary rather than a replacement of other calibration methods. Thus, we wanted to test this claim and investigate whether the combination would result in similar or improved calibration results.
>
> - **W4:** While we extended the related work section a bit, we decided to keep the focus on our empirical results and refer to other survey studies instead. However, upon request, we are willing to extend the section even further.
>
> - **W5:** We added the Brier score as a new evaluation metric to get new insights.
>
>
> In addition, we fixed the two minor changes you highlighted.
>
>
> We hope that we have changed the paper to your satisfaction and are looking forward to your answer.

---

### Review · Reviewer_w9zd · 2025-04-03

**Summary Of Contributions:**

The authors present a reproduction study of the paper “Decoupling Feature Extraction and Classification layers for Calibrated Networks”, referred to as ‘original’ henceforth. The original work proposes a two-stage training (TST) paradigm to improve calibration of (image-domain) neural networks. The original work also proposes a variational (V-TST) version.

The reproduction effort repeats the main experiments of the original work on 2 out of 3 datasets, and also repeats two ablation studies, namely applying the technique with a smaller, less complex model, and applying it with pre-trained image representations networks.

Furthermore, the authors add two ablation studies, namely on the depth of the classification network, and on the number of monta-carlo samples during training with V-TST.

Finally, the authors extend the work by experimenting with the addition of focal loss next to TST and V-TST.

**Audience:**

Yes

**Claims And Evidence:**

No

**Requested Changes:**

The main change I would like to see is for Table 1 and Table 2 to match closely with the original work, especially for Cifar100. In paragraph 5.1, the authors note the following:

| We noticed that our base models for CIFAR100 achieved the lowest validation loss and were obtained in the
early epochs, 18 and 16, even though the training accuracy reached around 100% and the validation accuracy
kept improving to similar levels reported in the original paper. Similar results may have been obtainable by
exploring more seeds or saving the model with the highest validation accuracy rather than the model with
the lowest validation loss.

Can the authors clarify whether their experiments use the base model with the lowest validation LOSS, and whether the original code uses base models with the lowest validation ACCURACY? I think this is a very relevant difference. It is usually the case that the validation accuracy can increase at the same as the validation loss, as the model increases in confidence during training, which leads to higher loss values on samples where the model was confidently wrong. At the same time, the validation accuracy still increases as there are things to learn. This has, of course, a large impact on calibration, as the longer you train, the less calibrated the network will be. Therefore, I can imagine that this small change in experimental methodology can lead to large effects in the calibration results.

**Strengths And Weaknesses:**

# Strengths

1. The paper has repeated the most relevant aspects of the original work.
2. The authors add relevant and interesting ablation studies.
3. The authors combine the original method with an interesting, relevant methodology in focal loss.

# Weaknesses

1. The authors do not reproduce the original findings to my satisfaction. It is not required for reproducibility studies to replicate the same results, but more steps could be attempted to reproduce exactly before having enough evidence that the results were not reproducible.

---

> ### Author Response · Authors · 2025-04-20
>
> We want to thank you for your feedback and time to review our submission. We have, to the best of our ability, tried to improve and change our submission in accordance with your feedback.
>
> To begin, we would like to clarify that we did use the lowest validation loss, just as in the original paper. Thus, we also found it surprising that we struggled that much to reproduce the results for CIFAR100. We did use the authors’ codebase and followed their instructions. It is especially odd as the other results could be reproduced with ease. In their paper, the original authors write:
> > “For all models trained we use early stopping based on the validation loss.“
>
> Thus, we followed the same procedure as the one reported in their paper.
>
> After further attempts, we managed to train a WRN-28-10 model with a different seed that matched the accuracy and overall trend of the original paper more closely. Unfortunately, the original paper did not specify which seed they ran to obtain their results. Thus, we had to try eight different seeds to obtain the results we have now. However, they still deviate from the original ones by some percentage points.
>
> While we get a lower accuracy on the test dataset, we would also like to stress that our model reached an accuracy of almost 100% during training and around 70% validation accuracy. However, this only happened later during training but most models had the lowest validation loss during the first epochs (between 15 and 30) with one (out of the eight) exception, where it reached the lowest validation loss in epoch 61. This is the new model that we included now instead of the one we previously had. We hope that we were able to explain all of this at the right level of detail in our updated version of the paper.
>
> As you already mentioned, the main goal is not to replicate the same exact results. However, we hope the obtained results are now more to your satisfaction. If you have any more suggestions on how we can try to obtain even more closely aligned results, we would be happy to attempt them.
>
> Thank you again for your time and consideration. We are looking forward to your answer.

---

### Review · Reviewer_fWJz · 2025-04-07

**Summary Of Contributions:**

The paper reproduces results from another paper which proposes a two-stage training procedure that decouples feature extraction and classification layers for better network calibration. Further, a few ablation studies have been conducted.

**Audience:**

No

**Claims And Evidence:**

Yes

**Requested Changes:**

I would suggest the authors, to shorten the reproduction of prior work or move it to the appendix.
The ablation studies require a more focused narrative; please clarify the key takeaways.
Furthermore, please incorporate the missing information detailed above.

**Strengths And Weaknesses:**

The paper's emphasis on reproducibility is valid and important.
However, the contribution, in its present form, seems better suited for a workshop setting than a full paper at TMLR.
Overall, the paper is missing an explanation of why additional settings were tested. The conducted ablations beyond the reproduction of prior work are missing a streamlined orientation e.g. in Section 5.2. Why did the authors use these additional models besides having different parameter counts?

Further, the results on CIFAR-100 are not competitive, with an accuracy of 64% also metrics such as ECE are not reliable anymore. However, a discussion on the reliability of ECE is missing.
Additionally, the authors fail to report what is actually used as confidence. The output after Softmax? If so, there is also a lot of literature discussing that the output after Softmax is not a suitable metric for confidence.

Tables are not referenced. Thus, it is not clear which Table belongs to which experiment.
Further, it is not clear to me what m and n a representing in Table 8.

Another question: Why is the 2-stage training with the focal loss so bad, can the authors elaborate more on that?

The authors omit to cite any of their data sources (CIFAR, ImageNet...)

The conclusion is too long and not precise enough.

---

> ### Author Response · Authors · 2025-04-20
>
> We want to thank you for your feedback and time to review our submission. We have, to the best of our ability, tried to improve and change our submission in accordance with your feedback.
>
> We changed the following points:
> - We added more motivation for why we conducted the ablation studies we chose (mostly in section 3.4). For example, we used ResNet-50 and EfficientNet because they are similar to and different from the WRN, respectively. Furthermore, the experiment sections now have a streamlined orientation and clearer explanations on what each experiment investigated.
>
> - We ran more seeds for the CIFAR100 results and obtained results that are more aligned with the original results. Additionally, we added a short discussion of the reliability of ECE (Appendix B) and included the Brier score to get a more comprehensive view.
>
> - We are aware that the output after softmax is not well calibrated and included some discussion about this as well. However, as the original paper was using the output after the softmax as the confidence, we also used it to ensure comparability.
>
> - We think we referenced all tables in our paper. Do you have an example where you feel like proper references are missing?
>
> - In Table 8, m represents the number of Monte Carlo samples during inference and n the number of MC samples during training. We added this explanation to the paper.
>
> - We tried to give a more thorough explanation on why 2-stage training with focal loss can also worsen calibration. We think it has to do with focal loss not being good in the second stage, where it has to deal with frozen features learnt from the first stage when retraining the last FC layers for further calibration. In this setting, CE loss seems more appropriate as focal loss focuses more on hard examples and aims to reduce overconfidence, which might not work well with having no flexibility in the feature extraction layers.
>
> - We added proper citations to the datasets, and made the conclusion section shorter and more concise, highlighting the key points. We also moved the tables with the reproduced ablation study results to the appendix.
>
> We hope that we have changed the paper to your satisfaction and are looking forward to your answer.

---

### Decision · Action_Editor_AVaZ · 2025-05-20

**Recommendation:** Accept with minor revision

**Comment:**

The authors have addressed some of the concerns raised by the reviewers. However, the two reviewers still have concerns regarding the justification for the method design and necessary explanations. The reviewers recommend the authors provide more explanations for performance drivers and ablation studies, and conduct more experiments to show the combination of TST with various calibration methods beyond focal loss in an online setting.

**Audience:**

Yes, some individuals in TMLR's audience will be interested in this paper as the calibration of neural networks is an important topic in deep learning.

**Claims And Evidence:**

The reproducibility study of this paper is valid and convincing.

**Resubmission Of Major Revision:**

The authors may consider submitting a major revision at a later time.